# KidSpeak: A General Multi-purpose LLM for Kids' Speech Recognition and Screening

## Abstract

With the rapid advancement of conversational and diffusion-based AI, there is a growing adoption of AI in educational services, ranging from grading and assessment tools to personalized learning systems that provide targeted support for students. However, this adaptability has yet to fully extend to the domain of children's speech, where existing models often fail due to their reliance on datasets designed for clear, articulate adult speech. Children, particularly those in early developmental stages or with speech and language pathologies, present unique challenges that current AI models and datasets are ill-equipped to handle. To address this, we introduce KidSpeak, a multi-task speech-enhanced Foundation Model capable of both generative and discriminative tasks specifically tailored to children's speech patterns. Our framework employs a two-stage training process that incorporates *phonetic knowledge* into the speech encoder, achieving an average accuracy of 87% across four separate tasks. Furthermore, recognizing the limitations of scalable human annotation and existing speech alignment tools, we propose the Flexible and Automatic Speech Aligner (FASA) and leverage the method to construct high quality datasets for training and evaluation. This novel alignment tool significantly improves the quality of aligned children's speech from noisy data, enhancing data quality by $13.6\times$ compared to human annotations, as demonstrated on the CHILDES dataset. To the best of our knowledge, KidSpeak and FASA represent the first comprehensive solution designed for speech and language therapy in children, offering both a multi-purpose speech LLM and a robust alignment tool. Code is available at Here.

## 1 Introduction

Humans begin to acquire the fundamental cues of vocal communication as early as **3 months of age** (USDHHS et al., 2017). As development advances, some individuals master their vocal abilities to such a degree that they are capable of vocalizing with over **1000 kHz**, allowing for the conveyance of complex and nuanced ideas and emotions Garnier et al. (2010)[1]. On the other hand, *hearing* begins as early as the **28th week of gestation** in humans (Querleu et al., 1988), eventually leading to an auditory capacity capable of discerning frequencies as precise as **0.5 Hz** within a range of 20 Hz to 20 kHz (Romand & Varela-Nieto, 2014). Despite these remarkable developmental milestones, numerous challenges persist in early speech acquisition. In fact, nearly **1 in 12 children** in the U.S. aged 3 to 17 has experienced a disorder affecting voice, speech, language, or swallowing, with almost half of them not receiving any intervention services in the past year (Black et al., 2015). These statistics highlight that, despite our advanced auditory and vocal capabilities, we continue to face significant barriers in the early diagnosis and treatment of speech-related disorders.

The setbacks are further compounded by the dearth of data pertaining to kids' speech and vocalizations, which poses additional challenges towards the development of automated computational tools. Consequently, the current state-of-the-art ASR systems remain lim-

---

[1]For reference, the normal adult male voice ranges from 90 to 155 Hz, while adult female voices range from 165 to 255 Hz (Fitch & Holbrook, 1970).

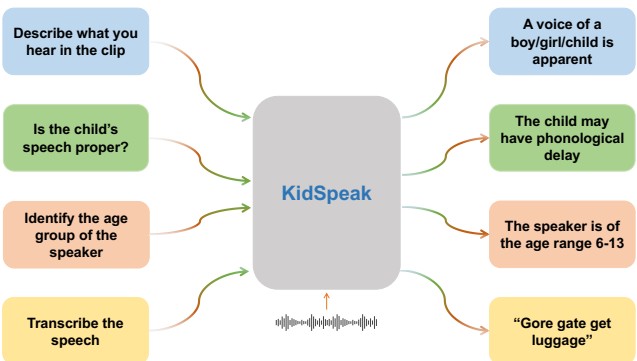

Figure 1: **Overview:** We propose KIDSPEAK, a multi-purpose speech based LLM aimed at diagnosis and transcription of kids' speech. The framework leverages a customized speech encoding procedure incorporating phonetic information enhancing downstream performance.

ited to the use of the widely popular datasets of LIBRISPEECH (Panayotov et al., 2015) and LIBRIVOX (Shankar et al., 2024), while also incorporating some of the lesser-known open-source datasets such as WSJ (Garofolo et al., 1993), CORAAL (Shankar et al., 2024), and TED-LIUM (Rousseau et al., 2012), along with their processed versions. Furthermore, the dataset creation phase itself for kids is burdened by the *labour-intensive nature* of the task, considering the nuanced speech and articulation patterns of children. This often leads to current approaches assuming proper pronunciation and articulation, **failing to account** sufficiently for kids' speech, and faltering *significantly more* with **accented and non-native kids' speech**, often generating *offensive and inaccurate transcriptions* (Ramesh et al., 2022). As an exemplar, the following transcription showcases a child's speech using the state-of-the-art ASR systems, WHISPER (Radford et al., 2023) and WAVE2VEC 2.0 (Baevski et al., 2020). The child is a **4-year-old non-native boy**.

> **Utterance**: *and they are looking at the frog; and because he cracked his egg*
> **Whisper**: *and they recognize the fog; and because do you grab this egg?*
> **Wav2Vec**: *unfated in that the fog; and because fee practis ed*

In an attempt to overcome these major hurdles, we introduce KIDSPEAK, a speech-based LLM with multi-task capacities of ASR, gender and dialect identification, and speech pathology classification, trained on a curated corpus of kids' speech through instruction tuning. The framework is based upon the foundations of spoken language understanding adapted towards kids' speech transcription and diagnosis of speech language pathologies. We train the method using a specialized two-stage procedure, wherein we utilize simultaneous phonetic and English transcription as a pre-training task for the WHISPER ASR model in order to incorporate *phonetically informed encoding* capacities into the encoder of WHISPER, as the first stage. Subsequently, the encoder of the model is used in the final framework.

Additionally, we acknowledge several limitations inherent in the existing datasets for children's speech. Given the **scarcity of relevant data**, we explore the CHILDES (MacWhinney, 2000b) corpus as a resource for children's speech. However, it is important to note that the transcriptions within this dataset are **significantly compromised**. The annotators involved are often engaged in multifaceted tasks, as the children included in the corpus frequently exhibit speech and language disabilities. Consequently, some annotators focus specifically on issues such as stuttering or speech sound disorders, while others address dialectical variations. This diversity in annotation purpose leads to **inconsistencies in human transcriptions**, limiting their applicability for developing robust automated systems. We therefore develop a new forced alignment tool FLEXIBLE AND AUTOMATIC SPEECH ALIGNER (FASA), allowing us to extract accurate, aligned, and well-segmented audio segments and the corresponding transcriptions under flexible conditions, creating a corpus for KIDSPEAK. **Our main contributions** are, ❶ We develop KIDSPEAK, a novel multi-task speech-based foundation language model aimed at diagnosis and transcription of children's

speech. ❷ We innovate a two-stage training procedure for the audio encoder, in order to incorporate phonetic information into the encoder, provably enhancing the downstream performance of the framework. ❸ We develop the FLEXIBLE AND AUTOMATIC SPEECH ALIGNER, a novel forced alignment tool, enabling extraction of accurate and aligned audio from noisy speech and demonstrate its utility over the CHILDES corpus in our framework.

## 2 RELATED WORK

Availability of data over the visual and descriptive-visual domain has spurred a preponderance of work towards understanding the visual domain. We witness a similar emergence in the aural understanding domain. Herein we provide an abridged summary of the contemporary work relevant to this manuscript. A more detailed description of the literature is provided in the Section A.1 under the Appendix.

**Speech-based LLMs and Spoken Language Understanding:** Challenges in encoding speech with LLMs stem from handling long sequences of audio. GSLM (Lakhotia et al., 2021), TWIST (Hassid et al., 2024), and SpeechGPT (Zhang et al., 2023) address them by using quantized speech representations using models such as HuBERT (Hsu et al., 2021). While others employ log-mel spectrograms to develop representations which are then combined with textual data for multi-modal generative tasks, such as speech recognition, generation and understanding (Fathullah et al., 2024; Nachmani et al., 2023; Zhao et al., 2023; Gong et al., 2023), using ASR models such as WHISPER (Radford et al., 2023), WAV2VEC (Baevski et al., 2020), Conformer (Gulati et al., 2020), and AST (Gong et al., 2021), or using multimodal retrieval based models such as ImageBind (Girdhar et al., 2023) like the PandaGPT (Su et al., 2023). *Our work innovates techniques essential for processing and understanding of kids' speech.*

**Kids' Speech:** A key limitation of current works is the insufficient handling of nuanced speech variations, such as accents, dialects, intonations, and developmental or disordered speech, as is typical in children. The field of children's speech recognition remains underresearched, with only a few notable approaches, such as LSTM-based disfluency detection (Venkatasubramaniam et al., 2023) and teacher-student models (Plantinga & Fosler-Lussier, 2019). *To the best of our knowledge, this work is the first to propose leveraging LLMs as multi-task models with diagnostic capabilities for children's speech, offering substantial potential in the domain of speech therapy and supporting Speech-Language Pathologists.*

**Forced-Alignment Toolkits** Traditional audio-transcription alignment relies on human annotators (Boersma & Weenink, 2007; Grover et al., 2020), which is not scalable for large datasets. While Kisler et al. (2017) offers components of a forced-alignment pipeline, it does not solve the alignment issue. Sheng et al. (2019) uses GANs for data augmentation in children's ASR datasets but does not introduce new data. Several studies utilize human-labeled transcriptions for forced-alignment (McAuliffe et al., 2017; Rodd et al., 2021; Zhang et al., 2023; Liu et al., 2023), with the Montreal Forced Aligner (MFA) being prominent (McAuliffe et al., 2017). However, MFA demands perfect alignment, limiting its effectiveness. *Our work improves over existing works and over human annotators by margins of over $13\times$.*

## 3 METHOD

We describe the method that we implement in order to create a multi-purpose speech LLM that exhibits potential as a useful diagnosis tool for speech-related impairments. The framework is trained using instruction finetuning. In summary, we utilize an audio encoder pre-trained using a targeted procedure, to generate representations which are subsequently post-processed and prepended to the textual embeddings and processed by a pre-trained LLM in order to generate answers. The main framework is illustrated in Figure 2. We employ the pre-trained Vicuna 7B model (Chiang et al., 2023) as our main LLM[2]. We note

---

[2]We choose Vicuna as it is one of the best LLMs we began with. Due to computational contraints, we did not try other strong pretrained LLMs. However, we believe they would perform similarly and our conclusions remain.

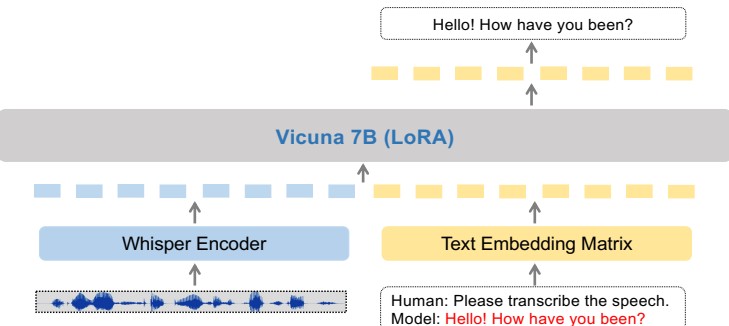

Figure 2: **The Proposed Framework:** KIDSPEAK uses phonetically informed speech features from the pre-trained multi-head WHISPER encoder. The features are concatenated with the text embeddings of the instructions during training endowing the framework with spoken context and textual instruction through self-attention.

that our data exhibits a significant domain gap from the pre-training, both in terms of format (audio vs text) and content (kids' speech). Therefore, in order to retain the general capacity of the LLM and avoid overfitting to the newer data, we finetune the LLM using Low Rank Approximation (LoRA) (Hu et al., 2021). This additionally helps with the memory footprint of the model, allowing for larger batch sizes.

### 3.1 SPEECH BASED LLM

Human: $< \texttt{Aud} > \mathbf{A} < /\texttt{Aud} >$
Human: $\mathbf{Y}_{q_1} < \texttt{STOP} >$ Assistant: $\mathbf{Y}_{a_1} < \texttt{STOP} >$
Human: $\mathbf{Y}_{q_2} < \texttt{STOP} >$ Assistant: $\mathbf{Y}_{a_2} < \texttt{STOP} > \ldots$

Figure 4: **Instruction Template:** We illustrate two instructions in the general input sequence that we implement to for the IFT procedure. The conversation structure comprises alternating exchanges between a human user and the KIDSPEAK, where tags $< \texttt{Aud} >$ and $< /\texttt{Aud} >$ demarcate the audio representations. The framework is trained to predict $\mathbf{Y}_{a_t}$ using the aural and instructional context. The $< \texttt{STOP} >$ is set to **###** in practice.

We employ a WHISPER-based encoder for speech in our main framework. The audio representations from WHISPER are prepended to the text embeddings consisting of instructions and the teacher-forced output. The complete sequence is further processed using the Vicuna LLM, incorporating self-attentive mechanism thereby incorporating audio-lingual context in order to learn and generate informed inference. However, we note that the native implementation of the WHISPER encoder generates encodings of shape $batch\_size \times 1500 \times 768$ leading to a significant increase in the memory footprint of the model due to the extensive sequence length and the consequent self-attention matrices. We therefore apply a post-processing step to the feature tensors produced by the WHISPER encoder. This step aims to reduce the final input sequence length whilst minimizing information loss. In this procedure, we aggregate multiple consecutive audio features (Figure 3) to form a cumulative representation that spans 80 milliseconds per feature vector. The aggregated features are then processed through a two-layer adapter network to align the feature space dimensions with those of the textual embeddings, which are further processed using the LLM. For each audio sample $i$, we create a multi-turn instruction following dataset $(\mathbf{Y}_{q_1}^{(i)}, \mathbf{Y}_{a_1}^{(i)} ... \mathbf{Y}_{q_T}^{(i)}, \mathbf{Y}_{a_T}^{(i)})$ illustrated in the Figure 4, wherein the instructions are randomly ordered during training. The framework is then trained using a conditional auto-regressive prediction objective

$$\underset{\theta_s, \theta_m}{\arg\min} \sum_{i=1}^{N} \sum_{j=1}^{T} \mathcal{L}_j^{(i)}, \text{with } \mathcal{L}_j^{(i)} = -\sum_{t=1}^{T_{a_j}^{(i)}} \log P\left(y_{a_{j,t}}^{(i)} \mid \quad y_{a_{j,<t}}^{(i)}, \mathbf{A}^{(i)}, \mathbf{Y}_{q_j}^{(i)}; \{\theta_{enc}, \theta_s, \theta_m\}\right),$$

for a sample indexed $i$ and the instruction $j$. The speech contexted LoRA parameters $\theta_s$ and the adapter MLP parameters $\theta_m$ are estimated, conditioning upon the speech sample $\mathbf{A}^{(i)}$ represented due to the frozen speech encoder parameters $\theta_{enc}$ and the instruction $\mathbf{Y}_{qj}$. We train the framework to predict the answer $\mathbf{Y}_{a_j}^{(i)}$ of length $T_{a_j}^{(i)}$. Additionally, we find that the a targeted encoding scheme for $\theta_{enc}$ benefits the framework, as we detail next.

## 3.2 MULTI-HEAD WHISPER PRETRAINING

A preponderance of developing speech is characterized by phonetic challenges wherein the child mispronounces similar phonetic units (Munson et al., 2012). Speech and language therapists must proficiently utilize phonetics for transcription to accurately diagnose and treat speech-sound disorders (Munson et al., 2012; Ball & Rahilly, 2002). Therefore, reliable pretraining in phonetic transcription is imperative, as inaccuracies can significantly affect clinical management and therapeutic outcomes. In recognition of these facets in diagnosis, we conduct a separate procedure in order to endow the speech encoder with phonetic information. The audio encoder employed is based on the WHISPER model which is designed to encode mono-channel audio sampled at 16 kHz, which is then transformed into log-Mel spectrogram images. The encoder generates audio features, with each feature tensor corresponding to a 20-millisecond segment of audio. For optimal performance, we utilize the default configuration, which processes audio in 30-second chunks, and affix it with separate dedicated decoders for the orthographic English transcription and phonetic transcription as illustrated in

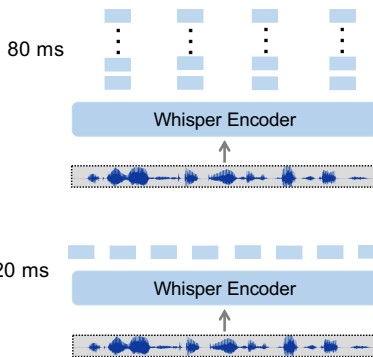

Figure 3: The speech embeddings are post-processed through a stacking mechanism (Top), ensuring adequate granularity. Thereafter the stacked features are projected onto the feature space of the LLM, ensuring synchronization between the two modalities.

Figure 5. This essentially yields a training regime wherein the same encoder facilitates both textual and phonetic transcriptions while the two decoders specialize in decoding the features into phonetic and English transcriptions respectively. The model is then trained using the next token prediction loss for both decodings simultaneously. The objective is,

$$\underset{\theta_{enc}, \theta_{dec}^{(en)}, \theta_{dec}^{(ph)}}{\arg\min} \sum_{i=1}^{N} \mathcal{L}_{ar}^{(i)}, \quad \text{where} \tag{1}$$

$$\mathcal{L}_{ar}^{(i)} = -\sum_{t=1}^{T_e^{(i)}} \log P\left(y_{t,e}^{(i)} \mid y_{<t,e}^{(i)}, \mathbf{A}^{(i)}; \{\theta_{enc}, \theta_{dec}^{(en)}\}\right) - \sum_{t=1}^{T_p^{(i)}} \log P\left(y_{t,p}^{(i)} \mid y_{<t,p}^{(i)}, \mathbf{A}^{(i)}; \{\theta_{enc}, \theta_{dec}^{(ph)}\}\right),$$

where $\mathcal{L}_{ar}^{(i)}$ is the combined auto-regressive loss for the $i$-th sample, $T_e^{(i)}$ and $T_p^{(i)}$ are the lengths of English and phonetic transcriptions, $y_{t,e}^{(i)}$ and $y_{t,p}^{(i)}$ are the target tokens, $y_{<t,e}^{(i)}$ and $y_{<t,p}^{(i)}$ are preceding tokens and $\mathbf{A}^{(i)}$ is the input audio. We specify the decoder parameters separately with $\theta_{dec}^{(en)}$ and $\theta_{dec}^{(ph)}$ representing the English and the phonetic decoders respectively using the same encoder parameters $\theta_{enc}$. We also utilize a special token |phn| to signify phonetic decoding. However, we hypothesize that explicit measures towards alignment of the two decoders may lead to enhanced generalization and understanding of the language through a unified feature space representation. Therefore, in order to enhance the alignment of the transcription mechanisms, we further process the downstream features of the decoders using two additional mechanisms.

**Contrastive Alignment** A cross-entropy-based contrastive loss (Chen et al., 2020) is incorporated, utilizing the |startoftranscript| token from both decoders to align pairs of English and phonetic sequences derived from the same audio more closely, while simultaneously separating sequences originating from different audio samples, formulated as,

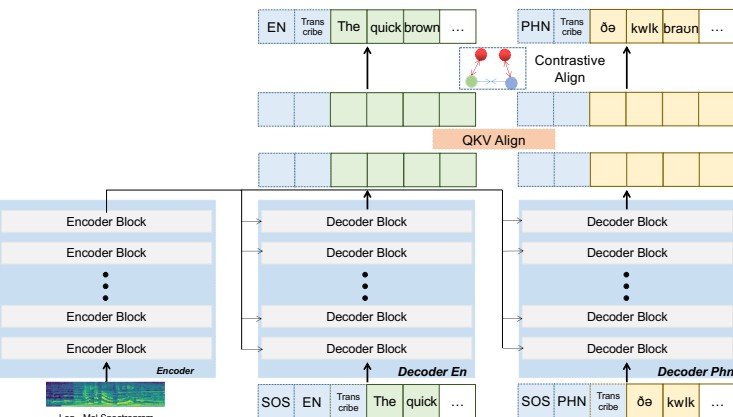

Figure 5: **Multi-head Whisper:** We employ two separate decoders to decode the same speech segment in English and its Phonetic counterpart. The decoders are further aligned using contrastive and cross-attentive mechanisms, synchronizing the procedure.

$$\mathcal{L}_{\text{con}}^{(i)} = -\log \frac{\exp\left(\text{sim}\left(\mathbf{h}_{\text{sos},e}^{(i)}, \mathbf{h}_{\text{sos},p}^{(i)}\right)/\tau\right)}{\sum_{j=1}^{B} \exp\left(\text{sim}\left(\mathbf{h}_{\text{sos},e}^{(i)}, \mathbf{h}_{\text{sos},p}^{(j)}\right)/\tau\right)}, \tag{2}$$

where $\mathcal{L}_{\text{contrastive}}^{(i)}$ represents the contrastive loss for the $i$-th sample, where $\mathbf{h}_{\text{sos},e}^{(i)}$ and $\mathbf{h}_{\text{sos},p}^{(i)}$ are the hidden states for the English and phonetic decoders, respectively, $\text{sim}(\cdot, \cdot)$ denotes the similarity function, $\tau$ is the temperature parameter, and the denominator sums over similarities over a batch of size $B$.

**Cross-attentive Alignment** This mechanism leverages cross-attention to synchronize the hidden states from two decoders derived from the same audio input. We implement the mechanism over the final hidden states of both decoders as formulated below,

$$\text{Residual}_p^{(i)} = \left(\text{Softmax}\left(\frac{(\mathbf{H}_e^{(i)}U)(\mathbf{H}_p^{(i)}U)^\top}{\sqrt{d_k}}\right)(\mathbf{H}_p^{(i)}U) \cdot D\right) + \mathbf{H}_p^{(i)}, \tag{3}$$

$$\text{Residual}_e^{(i)} = \left(\text{Softmax}\left(\frac{(\mathbf{H}_p^{(i)}U)(\mathbf{H}_e^{(i)}U)^\top}{\sqrt{d_k}}\right)(\mathbf{H}_e^{(i)}U) \cdot D\right) + \mathbf{H}_e^{(i)}, \tag{4}$$

where $\mathbf{H}_e^{(i)}$ and $\mathbf{H}_p^{(i)}$ represent the hidden states of English and phonetic transcriptions, $U(768 \times 1024)$ and $D(1024 \times 768)$ the upward and downward projection matrices, and $d_k$ the dimensionality of the keys. The upward projection enriches the representation space to capture detailed alignments, while the downward projection ensures the contextualized outputs are compatible with the original dimensions. In practice, we implement this mechanism using multi-head attention with 16 heads. The residuals are ultimately used in the evaluation of $\mathcal{L}_{ar}$ in Equation 1. A detailed justification for this scheme is provided under Section A.2.1. In summary, these alignment mechanisms ensure that phonetic and orthographic transcriptions are not only aligned but also mutually reinforcing, enhancing the encoder's ability to generate accurate and contextually relevant transcriptions for both modalities, ultimately improving the model's overall performance and utility in downstream tasks. The model is subsequently trained end-to-end using a linear combination of the two resulting loss functions given by $\mathcal{L}_{whisper}^{(i)} = \mathcal{L}_{ar}^{(i)} + \lambda \mathcal{L}_{con}^{(i)}$.

## 4 DATASET CONSTRUCTION

In this section, we present the FLEXIBLE AND AUTOMATIC SPEECH ALIGNER (FASA), a novel toolkit designed for forced alignment to create high-quality fine-tuning datasets. A

robust children's speech model necessitates a large, diverse dataset with accurately aligned audio and transcriptions. However, obtaining such high-quality data is challenging due to the distinctive speech patterns of children, particularly those with speech and language disorders. Human annotation is labor-intensive and requires domain expertise, as noted by Miller et al. (2016), with our experience showing that annotating a single audio segment can take *3-8 times longer* than its duration. Additionally, the quality of annotations varies significantly, especially in datasets like CHILDES (MacWhinney, 2000a), which cater to diverse transcription purposes, resulting in many transcriptions being incomplete or irrelevant. To enhance KIDSPEAK, a general-purpose forced alignment toolkit is crucial for extracting high-quality children's speech datasets from low-quality sources. Existing methods, such as MFA (McAuliffe et al., 2017), rely on accurate transcriptions, which are often impractical to obtain. Therefore, we propose a flexible and automated forced alignment toolkit that addresses various challenges in current children's speech datasets.

## 4.1 FASA DESIGN

Given a non-timestamped audio file and its noisy or incomplete transcription, forced alignment generates time-stamped audio segments paired with high-quality transcriptions. KID-SPEAK, like many modern automatic speech recognition systems, requires input audio to be divided into smaller segments during training. For instance, the WHISPER model (Radford et al., 2022) pads or trims audio inputs to 30 seconds. Consequently, when associating a non-timestamped transcription with a lengthy audio file, a forced-alignment toolkit is essential for creating a model-compatible dataset. Formally, the forced-alignment task involves an audio sample containing $n$ utterances, $\mathbf{A} := \{A_1, A_2, ...A_n\}$, and a transcription of $m$ "words," $T := \{T_1, T_2, ...T_m\}$. A "word" in $T$ represents a fundamental unit of transcription, which may refer to a sentence, a single word, or a phonetic symbol. The goal is to associate each $A_i$ with its corresponding words in $T$, from $T_{si}$ to $T_{ei}$, or indicate that $A_i$ lacks a transcription in $T$. We denote this association as $A_i = (T_{si}, T_{ei})$. A robust auto-alignment system should exhibit two crucial features. First, it must not assume that if $A_i = (T_{si}, T_{ei})$ and $A_j = (T_{sj}, T_{ej})$ with $i < j$, then $ei < sj$; an utterance appearing earlier in the audio does not guarantee its early appearance in the transcription. Second, $A_i$ may lack a corresponding $(T_{si}, T_{ei})$, implying that $A_i = \emptyset$. This means not all audio segments have transcriptions, and some audio may remain untranscribed. Similarly, $T_k \in T$ does not imply $T_k \in A$; not every word in the transcription corresponds to an audio segment. These features are essential as they relax the need for completeness and order in the provided transcription, mirroring more realistic scenarios. While a common method for obtaining large datasets of paired audio and transcriptions is through Internet scraping, many online transcriptions are noisy and incomplete, often with missing or misordered entries. Under these conditions, existing forced-alignment toolkits, such as those proposed by McAuliffe et al. (2017), are inadequate. Further details are discussed in Appendix A.1.

$$GT_k = \begin{cases} T_{A_k} = \{T_i, T_{i+1}, ..., T_j\}, & \text{if } T_{A_k} \in T \\ \emptyset, & \text{otherwise} \end{cases} \qquad (5)$$

## 4.2 WORKFLOW

FASA follows a five-module pipeline to automatically segment, label, and align a long audio file with its transcription, as illustrated in Figure 6. Among the five modules, the second and third are mandatory, whereas the other three are optional for enhancing the quality and quantity of the dataset. These five modules together maximize the correctness of forced alignment under flexible conditions. ① The first module applies a regular expression to clean the provided transcriptions and to exclude any non-alphanumeric characters. ② For the second module, modern ASR models will be used to obtain word-level timestamps of the transcriptions. Currently, sentence-level separations from the provided model are used as the segmentation marks for long audio. ③ After the second module, a folder consisting of audio segments and their corresponding predictions will be generated. The set of predictions for sentence-level utterances will be denoted as $\bar{T} = \{\bar{T}_1, \bar{T}_2, ..., \bar{T}_n\}$. For each utterance $A_k$, its predicted transcription will be $A_k = \bar{T}_{A_k}$. The third module will apply a sliding-window

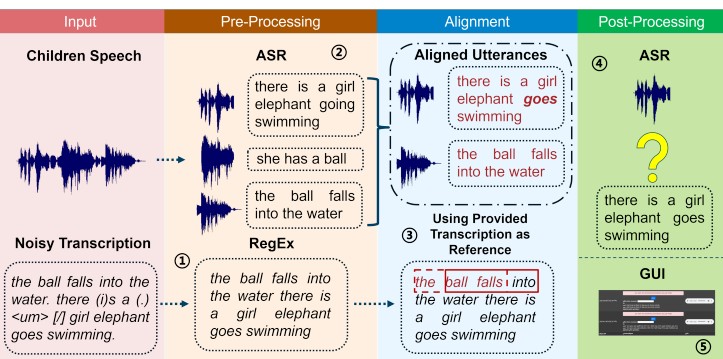

Figure 6: **Pipeline of FASA:** The input is an audio file and a transcription. Module ① optionally cleans the input transcription; module ② segments and makes predictions on the audio; module ③ forced-aligns audio segments with the provided transcription using Algorithm 1; module ④ performs post-generation checking (PGC); and module ⑤ allows user to augment dataset via manual selections. The entire system besides module ⑤ is automatic.

Algorithm 1 (in Appendix A.6) to find the best matching from the provided transcription ($T$) for each utterance ($A_k$). After this module, two datasets will be generated. The first dataset DATA$_{align}$ is what the algorithm finds close alignment between the prediction and provided transcription that is within a threshold. The second dataset DATA$_{verify}$ is what the algorithm finds slight mismatches between the prediction and the transcription. For DATA$_{align}$, the provided transcription from $T$ will be used as the ground truth of the utterance. ④ The fourth module, post-generation checking (PGC), is an optional module that iterates through DATA$_{align}$ to find if there are significant mismatches between a second-round prediction and the aligned transcription on sentence length. The implemented metric for PGC is based on the difference in sentence length between the results of a second-round prediction and the aligned transcription. If the difference is greater than a threshold, the utterance and its transcription will be removed from DATA$_{align}$. ⑤ The fifth module, user selection, is an optional module that launches a graphical-user-interface (GUI) that allows the user to listen to, select, or input correct transcription for each utterance in DATA$_{verify}$ so that they could be added to the dataset. After the two optional modules, FASA assumes the validity of DATA$_{align}$, which will be used as the final output dataset. Furthermore, FASA features additional qualitative and user-friendly traits as described in Section A.3.

Table 1: **Dataset and attribute specifications:** We present the attributes that we avail from the various dataset compiled to create the training corpus. **Dis.**: Disorder Labels, **Age**: Exact Ages of the speakers, **Gen.**: Gender attribute information, **Trans.**: Speech transcription for the audio.

| Dataset | Speakers | Utterances | Time | Dis. | Age | Gen. | Trans. |
|---|---|---|---|---|---|---|---|
| UPX | 20 | 2789 | 07:01:23 | ✓ | ✓ | ✓ | – |
| CSR | 11 | 639 | 00:26:58 | – | – | ✓ | ✓ |
| ENNI (FASA) | 352 | 4402 | 15:16:51 | – | ✓ | ✓ | ✓ |
| Clinical Eng (FASA) | 1540 | 59539 | 30:11:40 | – | ✓ | ✓ | ✓ |
| Clinical Other (FASA) | 292 | 59539 | 4:18:16 | – | ✓ | ✓ | ✓ |

### 4.3 DATASET SPECIFICATIONS

We compile multiple open-source datasets in order to test our framework against a wide variety of instructions, in addition to datasets generated by FASA leading to **over 57 hours of high-quality data**. The corpus was built in order to adequately represent children with speech pathologies and those with clear speech, and containing a rich collection of speaker related attributes that we aim our method to predict and generate. We summarize the data in Table 1. A broader description of the datasets is provided in Section A.4.

## 5 EVALUATION

We conduct training and performance evaluation of our method across several distinct tasks, each based on specific speech traits. These traits are extracted from the corresponding ground-truth labels available in the datasets referenced in Table 1. The tasks are described as follows:

- ◆ *Disorder Classification*: Speech disorders are categorized into the following classes: ❶ inconsistent phonological disorder, ❷ consistent phonological disorder, ❸ childhood apraxia of speech, ❹ phonological delay, ❺ vowel disorder, ❻ articulation disorder, and ❼ no disorder, based on the ground truths provided in the Ultraphonix (UPX) subset of the Ultrasuite repository Eshky et al. (2019). Detailed descriptions of these disorders are provided in Section A.5.

- ◆ *Gender Classification*: We conduct binary gender classification using the ground-truth gender labels available in the datasets. Predictions are made only where gender information is explicitly provided.

- ◆ *Age Group Classification*: Age labels are sourced from the ENNI dataset. To account for the minimal acoustic differences between closely aged children, we divide the age range into two groups: 1–5 years and 6–13 years.

- ◆ *Transcription*: We compile the transcriptions available for the kids with no speech-related disorders in order to create a reliable benchmark for training and evaluating speech recognition models. We train the framework to transcribe the speech and evaluate using the word error rate and character error rate metrics.

We evaluate the performance of the classification tasks using the accuracy of inference. Additional details for the configuration of the training setup are provided under Table 6.

**Phonetic Pre-training Enhances Speech Diagnosis and Transcription in Kids** In addition to garnering benefits over the KIDSPEAK framework, the aligned training procedure for the WHISPER model is beneficial for the transcription performance of the WHISPER model as shown in Table 2, where we evaluate the Phonetic Error Rate of the models using various configurations. The multi-task evaluation in Table 3 compares the performance of KIDSPEAK and KIDSPEAK (MH-WHISPER) in addition to the PandaGPT (Su et al., 2023), which we adapt to our application of children's speech diagnosis. While KIDSPEAK achieves a strong performance in gender classification and disorder classification, the MH-WHISPER variant shows notable improvements in disorder classification, word transcription, and character transcription accuracy. Additionally, both methods maintain high accuracy in age-group classification significantly overcoming the performance of PandaGPT. Overall, the MH-WHISPER demonstrates a significant enhancement in transcription and classification tasks compared to the original

Table 2: **Whisper MH comparison**. We evaluate the Phonetic Error Rate (PER) for our scheme comparing against fine-tuned open-source ASR models and ablate the alignment objective over the TIMIT dataset (Garofolo, 1993). (MH-1: WHISPER with two decoders; MH-2: MH-1 + Cross Attn.; MH-3: MH-2 + Contrastive Alignment)

| Method | PER |
|---|---|
| WHISPER | 10.1 |
| WAV2VEC 2.0 | 9.7 |
| WHISPER MH-1 | 9.6 |
| WHISPER MH-2 | 9.2 |
| WHISPER MH-3 | **8.6** |

KIDSPEAK. The performance of the MH-WHISPER model, which integrates phonetic and English data, underscores the critical role of phonetic knowledge in tasks related to children's speech. Children often exhibit unique speech patterns, including phonological disorders and developmental variances. By incorporating phonetic information, the model better understands these nuances, allowing it to differentiate subtle pronunciation variations common among young speakers. This phonetic grounding enhances the model's ability to generalize across diverse dialects and individual speech patterns, ultimately contributing to more reliable assessments and interventions in speech-related applications for children.

Table 3: **Multi-Task Evaluation:** We present the evaluations through mean scores over three separate runs for the tasks. $a_{\pm b}$ notation represents mean and standard error of the runs. **GCA:** Gender Classification Acc., **DCA:** Disorder Classification Acc., **WTA:** Word Transcription Acc., **CTA:** Character Transcription Acc., **ACA:** Age-group Classification Acc. (**WTA** = 1 - Word Error Rate and **CTA** = 1 - Character Error Rate).

| Method | GCA | DCA | WTA | CTA | ACA | Average |
|---|---|---|---|---|---|---|
| PandaGPT | $61.0_{\pm 0.3}$ | $42.6_{\pm 4.1}$ | $6.6_{\pm 1.9}$ | $13.6_{\pm 1.2}$ | $84.3_{\pm 0.4}$ | $50.3_{\pm 2.3}$ |
| KidSpeak | $73.8_{\pm 0.4}$ | $85.0_{\pm 2.9}$ | $82.2_{\pm 0.4}$ | $87.0_{\pm 0.3}$ | $93.8_{\pm 0.5}$ | $84.4_{\pm 0.9}$ |
| KidSpeak (MH-Whisper) | $73.3_{\pm 0.1}$ | $88.8_{\pm 2.3}$ | $87.8_{\pm 0.2}$ | $91.0_{\pm 0.2}$ | $94.1_{\pm 0.1}$ | $\mathbf{87.0}_{\pm 0.6}$ |

Table 4: Manual inspections on the generation quality of FASA on two randomly selected audio files and their transcriptions. **AU:** Aligned Utterances, **AW:** Aligned Words.

| Model | AU | AU Error (%) | (AW) | AW Error (%) |
|---|---|---|---|---|
| MFA | 17 | 16 (94.12%) | 1524 | 1523 (99.93%) |
| FASA | 81 | 1 (**1.23%**) | 903 | 2 (**0.22%**) |

**FASA alignment Outperforms Human Annotators**  Considering the vast size of the dataset, we randomly selected two audio files and transcriptions from the 352 recordings in the ENNI dataset, and report the manual inspection results for data generated by FASA with these files in Table 4. Several results are worthy of emphasizing here. First, since the transcriptions are noisy, MFA (McAuliffe et al., 2017) completely fails to properly align the audio segments with the correct transcription. [3] To be specific, both documents have missing transcriptions corresponding to the beginning of the audio, which results in 99.93% AW Error. This is because MFA tries to align all the words from the beginning, but since those words do not have available transcriptions, the entire system fails. Second, FASA incorrectly aligns one utterance with its transcription. For that utterance, it misses the "so the" sound at the end of the utterance, and the two words are not recorded into the aligned transcription. Manual inspection finds that the speaker stuttered and repeated "so the", which might be the issue of the model not picking up that trailing sound in the segmented utterance. Lastly, FASA's result is potentially much better than human annotators. Attia et al. (2023) reports that 5 out of 393 hours of speech in MyST dataset (Pradhan et al., 2023) are potentially incorrect with WER> 50%, resulting in **3%** increase in WER for the entire training dataset. Compared to human annotators that were used to annotate MyST, FASA achieves one magnitude lower WER (**13.6×**) without requiring any human labor.

## 6 Conclusion

In conclusion, this work presents significant advancements in the field of children's speech analysis. First, we introduce KidSpeak, a pioneering multi-task speech-based foundation language model designed specifically for diagnostic tasks pertaining to children's speech. Second, we propose an innovative two-stage training procedure for the audio encoder that effectively integrates phonetic information, leading to marked improvements in both diagnostic and transcription performance, as well as downstream performances when integrated with KidSpeak. Finally, we develop the Flexible and Automatic Speech Aligner, a novel forced alignment tool that extracts accurate and aligned audio from noisy speech, allowing us to create a high quality kids' data corpus. Collectively, these contributions enhance the capabilities of speech analysis frameworks for children, paving the way for future research and applications in this critical area.

---

[3]This is not to say that MFA is not a good model. MFA works fine with high-quality transcriptions that are an approximate match of the audio. However, if the audio/transcription match before the alignment is not good, MFA will not produce anything meaningful.

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

# A  APPENDIX

| | |
|---|---|
| 9y old Girl: | *"The frog jumped off the box."* |
| User: | Determine the gender of the speaker in this clip. |
| KIDSPEAK: | The child in the clip is a girl. |
| | |
| 4y old Boy: | *"The elephant grabs it and drops it in the water."* |
| User: | Transcribe the spoken words into text. |
| KIDSPEAK: | The elephant grabs it and drops it in the water. |
| | |
| 7y old Boy: | *"Just turn the tap on for water."* |
| User: | Evaluate whether the speech in this clip is typical or impaired. |
| KIDSPEAK: | The manner of speaking seems natural. |
| | |
| 10y old Girl: | *"The hippo jumps in the pool."* |
| User: | Estimate the age-group of the speaker from this clip. |
| KIDSPEAK: | It is estimated that the speaker may be between 6 - 13 years old. |

Table 5: Conversation with KIDSPEAK.

## A.1  RELATED WORK

**Audio-Visual Understanding:** Contemporary work has attained remarkable feats with regards to understanding the visual and lingual modalities, often in synergy with each other, owing to the large scale availability of datasets facilitating research. Several works including but not limited to LLaVA (Liu et al., 2024), PaLME (Driess et al., 2023), Flamingo (Alayrac et al., 2022), BLIP (Li et al., 2022) and GPT-4 (Achiam et al., 2023) incorporate instruction based tuning in order to incorporate multi-modal capabilities in language models. The training routines for these methods involve tokenization of the image modality allowing for the formation of a sequence which one may incorporate into instruction/text token sequences using self attention (LLaVA, PaLME), cross attention (Flamingo) or through a separate network (BLIP). This often consists of multiple stages wherein the initial set of stages are aimed at priming the learnable parameters for the newer modality. In view of their tremendous potential, we also witness applications of these advancements helping significantly enhance human-AI interaction by improving search engines, supporting creative tasks, and, importantly, advancing accessibility, particularly for the image modality. For instance, these technologies can be utilized to enhance accessibility tools, such as through augmented communication (Chanjaradwichai et al., 2019), as assistive learning tools (Padmanabha et al., 2024; Kazemitabaar et al., 2024) and sign language recognition systems (Gong et al., 2024).

**Speech based LLMs and Spoken Language Understanding:** Encoding speech using LLMs is faced with challenges associated with encoding very large sequences of aural representations, given that a 16kHz sampling of one second of audio contains 16000 unique representations of the medium in the frequency domain. However, several recent contributions quantized representations of speech provided by the HuBERT (Hsu et al., 2021), as leveraged by the works of Generative Speech Language Modeling (Lakhotia et al., 2021), TWIST (Hassid et al., 2024) and SpeechGPT (Zhang et al., 2023) wherein based on the applications, the representations are used in transformer (Vaswani, 2017) based encoder decoder systems or are combined with the textual embeddings in order to construct an interactive multi-modal system using instruction based tuning, with generative capabilities in audio. Another encoding scheme for audio is the use of log-mel spectrograms, processed

using encoders such as WHISPER (Radford et al., 2023), WAV2VEC Baevski et al. (2020), Conformer (Gulati et al., 2020) and the AST (Gong et al., 2021), in order to generate discrete representations which may similarly be combined with textual representations for understanding and generative tasks. Fathullah et al. (2024) incorporate the conformer architecture in order to enable speech recognition in LLMs. Spectron (Nachmani et al., 2023) uses the conformer encoder and splits the aural representations into prompt and continuation, enabling the continuation of aural speech and language using prompt during inference. Zhao et al. (2023) use the WavLM (Chen et al., 2022) and the WAV2VEC encoders whereas Gong et al. (2023); Ghosh et al. (2024) utilize the AST model and develop a question answering based training procedure to inculcate understanding in a LLM using the aural representations.

**Kids' Speech:** A predominant drawback of the concurrent works is the lack of support for nuanced speech based on a variety of accents, dialects and intonations, or developing or disordered speech as often characterized by kids. This challenge is further exacerbated by the relative lack of attention and research in the field of children's speech recognition and correction, resulting in a limited number of robust solutions tailored specifically to these needs. Liao et al. (2015) use a LSTM (Hochreiter & Schmidhuber, 1997) and CLDNN (Sainath et al., 2015) based architectures towards transcription and reduction of offensive generations. Plantinga & Fosler-Lussier (2019) use a GRU (Cho, 2014) based architecture with alignment loss to discourage generation during silence and a teacher student loss in order to improve transcription performance. Ramesh et al. (2022) leverage a masked word prediction based cloze task inspired by BERT based encoder systems in order to correct offensively transcribed speech by existing ASR systems. (Venkatasubramaniam et al., 2023) develop an LSTM based disfluency detection and classification architecture over an existing ASR system in order to enhance transcription. However, to the best of our knowledge, this is the first work that proposes utilizing a large language model (LLM) as a multi-task model with diagnostic capabilities within the speech domain. *We anticipate that this represents a promising avenue for future research, offering substantial benefits in the realm of speech therapy for kids and complementing the extensive efforts of Speech-Language Pathologists (SLPs).*

**Forced-Alignment Toolkits:** Traditionally yet still prevalently, alignment between the audio and its transcription is done via human annotators on various software (Boersma & Weenink, 2007; Grover et al., 2020). However, as discussed earlier, such practice is not scalable for large datasets. Kisler et al. (2017) contains some parts of a complete forced-alignment pipeline, but it does not address the fundamental problem of aligning audio with its transcription. Sheng et al. (2019) uses generative-adversarial networks (GAN) to perform data augmentation on children ASR dataset, but their work does not introduce diverse *new* data to the field. Recently, the Talkbank project announced its data processing pipeline that converts raw audio into CLAN-annotated transcriptions (Liu et al., 2023). While their work uses a similar backbone structure as ours, their complete pipeline relies on transcription generated by ASR models, whereas we faithfully adhere to the provided transcription as the ground truth. Thus, on downstream tasks such as fine-tuning ASR models, our dataset will be more usable because datasets generated by ASR models might cause severe degradation according to (Radford et al., 2022). On the other hand, there have been several works on forced-alignment ASR datasets with the assistance of human-labeled transcriptions (McAuliffe et al., 2017; Rodd et al., 2021; Zhang et al., 2023; Liu et al., 2023), with Montreal-Forced-Aligner (MFA) being the most popular toolkit (McAuliffe et al., 2017). MFA incorporates Kaldi (Povey et al., 2011) as the backbone, which uses the Gaussian Mixture Model (GMM) for its transcription generation process. However, while MFA works well with carefully annotated transcriptions, it requires the transcription to have a perfect match with the audio. That is, $A_1 \rightarrow \{T_1, T_2, ..., T_i\}$, $A_2 \rightarrow \{T_{i+1}, T_{i+2}, ..., T_j\}$, and so forth. Liu et al. (2023) faces similar issues that it lacks global matching ability between audio and transcription, hindering its usage in some subsets of the speech corpus. Moreover, while recent multi-modal large language models (MLLM) might have the potential of automating the alignment process (Zhang et al., 2023), they are much more resource-intensive compared to specific ASR models.

## A.2 Phonetically Endowed Multihead Whisper

### A.2.1 Decoder Alignment in Multi-head Whisper

The two alignment mechanisms address a critical need for alignment between phonetic and orthographic transcriptions within multi-modal speech processing systems. This alignment is essential for several reasons:

- ◆ *Phonetic and Orthographic Consistency*: Phonetic transcriptions represent the pronunciation of words, focusing on sounds, while orthographic transcriptions represent the written form of language. Aligning these two modalities ensures that the pronunciation (phonetics) and spelling (orthographics) are consistent with each other. This consistency is crucial for tasks such as speech recognition and language learning, where accurate mapping between spoken and written forms is required.
- ◆ *Enhanced Encoder Utility*: By aligning phonetic and orthographic transcriptions through the shared encoder, the model benefits from enriched feature representations. The encoder, which is common to both decoders, learns to produce more comprehensive and phonetically aware representations. This shared learning helps the encoder capture nuances that prove to be critical for improved understanding of pronunciation based nuances necessary for improved diagnostic capacities.
- ◆ *Robust Multi-Modal Learning*: Aligning the hidden states of phonetic and orthographic decoders allows the system to leverage complementary information from both transcriptions. Phonetics provides insights into pronunciation nuances, while orthographics offer context about spelling and grammar. The combined insights from both modalities lead to a more robust and versatile model capable of handling diverse linguistic tasks.

In the following, we demonstrate the phonetic capacities of our scheme using comparisons with Whisper and Wav2Vec trained over TIMIT. The phonetic alphabet used natively by TIMIT is illustrated here for selected samples. We notice that all of the models capture a meaningful pronunciation for each word of the examples listed. However, the targeted alignment scheme of our method captures the nuances in pronunciation, audible in the ground truth phonetic captioning, enabling a more accurate transcription and henceforth, better encoder representations, for therapeutic downstream tasks.

---

**1. SCRIPT: She had your dark suit in greasy wash water all year**

**Groundtruth:** shih hhehjh jhih **pau** dahk suw n **pau** grishih waash waadxer aal yiher
**Wav2Vec:** shix hvehjh jhuh **pau** dahk suxq en **pau** grisix waosh waodxax aol yihaxr
**Whisper:** shix hvehjh jhih **pau** dahk suxq en **pau** grisxix waosh waodxaxr aol yihaxr
**Whisper ours:** shih hhehjh jhih **pau** dahk suw n **pau** grishih waash waadxer aal yiher

---

**2. SCRIPT: Don't ask me to carry an oily rag like that**

**Groundtruth:** down aes **pau** my th **pau** kehriy ihn oylih raeg lay dhae
**Wav2Vec:** down aes **pau** my **pau** tx **pau** kehriy ixn qoyliy raeg lay dhae
**Whisper:** down aes **pau** my **pau** tx **pau** kehriy ixn qoylih raeg lay dhae
**Whisper ours:** down aes **pau** my th **pau** kehriy ihn oylih raeg lay dhae

---

### A.2.2 Disorder detection

In the following, we demonstrate the capacity of the phonetically endowed Whisper model using speech from the Ultraphonix dataset Eshky et al. (2019). The Whisper model was trained using the Multihead alignment scheme described in Section 3.2 using the TIMIT

corpus (Garofolo, 1993). Subsequently, we conduct inference using the phonetic decoder of the model over the speech of a 4 year old boy undergoing therapy for phonological disorder. The child mistakenly uses the sound of *"da"* in place of *"ga"* for words. For instance, the pronunciation for the word "luggage" (**"LUG-ij"**) and "gore" (**"GOw-ar"**) here are made as *"LAD-ij"* and *"DOw-ar"*. However, post therapy, the child learns to pronounce clearly as is captured by our model. We use the TIMIT phonetic transcription code here. **pau** indicates a pause in speech.

---

**SCRIPT:** gore gate get luggage

**Instructor:** g ow axr **pau** g ey t **pau** g eh t **pau** l ah g ux jh
**Child Pre:** d ow ax **pau** d iy t **pau** d ae t **pau** l ah d ix jh
**Child Post:** g ow aa **pau** g ih g eh ix t **pau** g ae t **pau** l ah g ih jh

---

As is evinced in the illustration, the phonetically endowed WHISPER correctly detects the improvements in pronunciation in **pre-** vs **post-** therapy of the child, thereby allowing for tailored features for targeted therapy based downstream tasks such as those implemented in KIDSPEAK.

### A.3 FEATURES OF FASA

Similar to existing auto-alignment toolkits, FASA requires an audio file and its corresponding transcription. However, due to high uncertainty in the raw dataset, FASA assumes only a minimal input format and does not require the transcription to be accurate. The ground truth (GT) for an utterance $A_k$ is defined by Equation 5 in the FASA pipeline. In contrast to previous forced-alignment toolkits, FASA deliberately ignores utterances without valid transcriptions, thereby enhancing quality.

FASA also incorporates beneficial design elements from established toolkits to enhance user convenience, following the same design principles as MFA. Users need only to place the audio file and its transcriptions in a designated folder before executing the program, after which all processes are fully automated, enhancing the user experience. FASA allows users to select and manually input transcriptions for utterances when the provided transcriptions are suspected to be inaccurate, ensuring precision and user control. Additionally, FASA features an optional post-generation check to automatically exclude incorrect alignments, minimizing errors from the underlying model.

### A.4 DATASETS

The Core-Ultraphonix (UPX) subset of the Ultrasuite repository (Eshky et al., 2019) provided the labels for speech with pathologies. Additionally, we incorporate the Children's Speech Recording (CSR) dataset by Kennedy et al. (2017). With FASA, we convert subsets of the Child Language Data Exchange System (CHILDES) (MacWhinney, 2000a) that contain English children's speech. Specifically, we use a collection of 352 children from ages 4 to 9. The children are performing the Edmonton Narrative Norms Instrument (ENNI) test (Schneider et al., 2005). To the best of our knowledge, this generated dataset will be the first at-scale high-quality dataset for young children from clinical recordings that is fully compatible with modern DL systems. While we only use the subset from CHILDES with rich clinical information for KIDSPEAK, FASA remains a generic forced-alignment toolkit that can extract many more datasets than the one used.

### A.5 SPEECH DISORDER CLASSES

The following provides a broad explanation of various speech-related disorders, along with seminal and intriguing citations in the field of speech-language pathology that KIDSPEAK is capable of diagnosing based on the speech patterns exhibited by the child.

◆ *Inconsistent Phonological Disorder*: This pediatric speech sound disorder is characterized by the inconsistent production of the same words across repeated trials

(Dodd et al., 2024). For example, a child may say *"bat," "gat,"* and *"at"* instead of *"cat,"* or produce *"log"* for *"dog"* one day and *"fog"* the next. Moreover, a child may say *"fider," "sider,"* and *"pider"* when attempting to pronounce *"spider"* (Dodd & Crosbie, 2010; Carter et al., 2019).

◆ *Consistent Phonological Disorder*: In contrast, this disorder is marked by the child's ability to produce the same errors consistently when attempting to articulate the same word. For instance, a child may reliably say *"tup"* instead of *"cup"* or *"wabbit"* for *"rabbit."* Such patterns indicate a stable phonological processing issue, as the child consistently makes the same substitutions or distortions (Felsenfeld et al., 1995; Bleile, 2002).

◆ *Phonological Delay*: This specific speech sound disorder entails developmental phonological errors that align with typical speech development patterns but persist longer than expected, often for six months or more, which can impact clarity and sound production (Orsolini et al., 2001). Children acquire speech by learning entire words rather than individual sounds; as their speech matures, they categorize words by their components, often simplifying sounds or sequences into easier alternatives (e.g., saying *"ca"* for *"cat"*) (Waring et al., 2022).

◆ *Vowel Disorder*: Vowel disorders are marked by difficulties in the positioning and sequencing of the articulators, particularly the tongue and lips, affecting vowel quality and accuracy. Incorrect positioning can lead to issues with vowel production, such as excessively long vowels or distortions. For instance, vowels may be partially voiced due to challenges in controlling vocal fold vibration, or they may exhibit excessive nasality from difficulties managing velopharyngeal closure. These spatial, temporal, and coordination difficulties often result in challenges in vowel production (Gibbon & Beck, 2002; Ball & Gibbon, 2002). Additionally, children may struggle with vowel lengthening or shortening, such as elongating the vowel in *"see"* for *"sit"* or shortening it in *"cat"* as *"kit,"* with omissions occurring as well (e.g., saying *"bll"* instead of *"ball"*) (Stoel-Gammon & Pollock, 2008).

◆ *Articulation Disorder*: This type of speech sound disorder is characterized by difficulties in accurately producing speech sounds due to the imprecise use of the lips, tongue, or throat. Individuals may demonstrate various symptoms, including the omission of sounds (e.g., final consonants), distortion of sounds (e.g., producing an *"s"* sound with a whistle), and challenges in coordinating the movements of their lips, tongue, teeth, palate, and lungs (Hall & Tomblin, 1978; Rvachew & Jamieson, 1989).

◆ *Childhood Apraxia of Speech*: Childhood apraxia of speech (CAS) is a neurological speech sound disorder characterized by impaired precision and consistency of movements underlying speech, absent neuromuscular deficits (e.g., abnormal reflexes or tone) (Davis et al., 1998; Association et al., 2007; Kummer et al., 2007). Children with CAS may encounter difficulties in speech production, such as trouble transitioning smoothly between sounds and syllables, groping movements of the jaw, lips, or tongue, vowel distortions, incorrect stress patterns (e.g., pronouncing *"banana"* as *"BUH-nan-uh,"*) equal emphasis on all syllables (e.g., saying *"BUH-NAN-UH,"*) separation of syllables with pauses, inconsistency in errors when repeating words, and voicing errors (e.g., saying *"down"* instead of *"town"*) (Carter et al., 2019).

### A.6 FASA WORKFLOW PSEUDOCODE

Here, we provide the pseudocode for the third module of FASA's workflow in Algorithm 1. FASA uses a sliding window algorithm with two thresholds to determine the final two subsets of audio segments. In the algorithm, $DIS$ is the Levenshtein distance between two sentences.

---

**Algorithm 1** sliding window to find the best matching

---

Input:     $A, T = \{T_1, ...T_m\}$, $\bar{T}$, alignment threshold $\sigma_a$, inclusion threshold $\sigma_i$.

Step 1:   Initialize holder for dataset of aligned segments: $\text{DATA}_{align} = []$
           Initialize holder for questionable segments: $\text{DATA}_{verify} = []$

Step 2:   **for** $A_k \in A$ **do**
           Get $A_k$'s transcription: $\bar{T_{A_k}} = \{\bar{T_i}...\bar{T_j}\} \in \bar{T}$
           Initialize minimum distance $D_{min} = \infty$, best starting index $BEST_i$, best
    length $BEST_l$
             **for** $a = 1, 2, \ldots, m$ **do**
               **for** $b = 1, 2, \ldots, (j - i)$ **do**
                  **if** $DIS(\bar{T_{A_k}}, T[a : a + b + 1]) < D_{min}$ **then**
                    $D_{min} = DIS(\bar{T_{A_k}}, T[a : a + b + 1])$
                    $BEST_i = a$
                    $BEST_l = b + 1$
                **end if**
               **end for**
             **end for**

Step 3:      let $GT_k = T[BEST_i : BEST_i + BEST_l]$
            **if** $WER(GT_k, \bar{T_{A_k}}) < \sigma_i$ **then**
               **if** $WER(GT_k, \bar{T_{A_k}}) < \sigma_a$ **then**
                  append $(A_k, GT_k)$ to $\text{DATA}_{align}$
               **else**
                  append $(A_k, GT_k, \bar{T_{A_k}})$ to $\text{DATA}_{verify}$
               **end if**
             **end if**
           **end for**

Output: $\text{DATA}_{align}$, $\text{DATA}_{verify}$

---

## A.7   Configuration

Table 6: Training setup for the methods.

| Attribute | PandaGPT | KidSpeak | MH-Whisper |
|---|---|---|---|
| Peak learning rate | 5e-5 | 5e-5 | 1e-4 |
| Batch size | 64 | 64 | 4 |
| Accumulate Steps | 8 | 8 | – |
| Max length | 512 | 512 | 448 |
| LoRA rank | 16 | 16 | – |
| LoRA alpha | 32 | 32 | – |
| Training steps | 10000 | 10000 | 63525 |
| Trainable parameters | 16.78M | 16.78M | 553.1M |
| Training device | 4*A6000 | 4*A600 | 1xA5000 |

