# OpenReview forum: "KidSpeak: A General Multi-Purpose LLM for Kids' Speech Recognition and Screening"
_ICLR.cc/2025/Conference — ICLR 2025 Conference Withdrawn Submission_

### Official Review · Reviewer_xWNH · 2024-10-28

**Soundness:** 3
**Presentation:** 3
**Contribution:** 3
**Rating:** 5
**Confidence:** 5

**Summary:**

This paper introduces KidSpeak, a speech-LLM that trains on both generative and discriminative tasks for children’s
speech. Additionally, the paper proposes the Flexible and Automatic Speech Aligner (FASA) to construct high quality datasets for training and evaluating KidSpeak. The novel alignment tool significantly improves the quality of aligned children’s speech from noisy data, enhancing data quality by 13.6× compared to human annotations.

The proposed KidSpeak and FASA represent the first comprehensive solution designed for child speech and language modeling using large language models (LLM).

**Strengths:**

The strengths of the paper are:

1. The paper is well organized and written.
2. The contribution is significant in the area of child speech and language modeling as the data quality of child speech corpus is always low. The proposed alignment tool can greatly boost the development of better models for child speech.
3. The paper is the first to use speech-LLM for child speech and proposed to train whisper-MH (multi-decoder whisper) to enhance the phonetic information in the whisper encoder.

**Weaknesses:**

The weaknesses of the paper are:

1. Insufficient experiments. The experiment section is far shorter than other sections. The current experiments seem to be not sufficient to support the claims made in the paper.

2. Missing references. In related work (Kids’ Speech), the authors is talking about child ASR remains under research. In fact, there are many papers recently are using speech foundation models for child speech recognition, following the ASR system development. e.g.

[1] Ruchao Fan, Natarajan Balaji Shankar, and Abeer Alwan. "Benchmarking Children's ASR with Supervised and Self-supervised Speech Foundation Models", Proc. Interspeech 2024, 5173-5177, https://doi.org/10.21437/Interspeech.2024-1353

[2] Jain, R., Barcovschi, A., Yiwere, M., Corcoran, P., Cucu, H. (2023) Adaptation of Whisper models to child speech recognition. Proc. INTERSPEECH 2023, 5242-5246, doi: 10.21437/Interspeech.2023-935

[3] R. Fan, Y. Zhu, J. Wang, and A. Alwan, "Towards Better Domain Adaptation for Self-supervised Models: A Case Study of Child ASR," in IEEE Journal of Selected Topics in Signal Processing, vol. 16, no.6, pp. 1242-1252, Oct. 2022, doi: 10.1109/JSTSP.2022.3200910

... few more

Speech-LLM papers:
Qwen2-audio, ...

**Questions:**

The paper is well organized, and the contributions are significant. However, the paper seems to lack comparisons to better support the claims in the paper. Here are my concerns and suggestions to the paper.

1. Is the KidSpeak trained with dataset described in Table 1? If so, I would suggest an ablation that training KidSpeak with the same dataset but pre-processed with MFA instead of the proposed FASA. The manual inspections on the generation quality of FASA in Table 4 look good. However, a direct comparison of performance on the KidSpeak model w/ and w/o FASA can better motivate the proposed FASA framework and show its significance.

2. In table 3, what is the performance of whisper model that is used to initialize the KidSpeak training? It is interesting to what is the gap of whisper model (S2S structure) and speech-llm (decoder-only structure). For other tasks, like gender classification, the whisper + LLM can serve as baselines for comaprions.

3. How would the KidSpeak compare to other methods on public child speech benchmark, like MyST? I understand that the kidspeak is trained with very limited instruction-following child speech data. However, the speech encoder and LLM backbone are well pretrained. If not using MyST for training, it is also interesting to see whether a very small amount of child speech data can train a good speech-LLM model with only adapters tunable.

4. How would the KidSpeak compare to other open-sourced speech-LLM models, like Qwen2-audio on the child speech tasks? The results of some open-sourced speech-LLM models can serve as additional baselines to emphasize the importance of developing child-specific speech-LLM.

5. Would the FASA be open-sourced?

The paper would provide significant contributions to the community if the experiment section were enhanced.

---

### Official Review · Reviewer_aDk1 · 2024-11-03

**Soundness:** 2
**Presentation:** 3
**Contribution:** 2
**Rating:** 5
**Confidence:** 4

**Summary:**

This paper introduces KidSpeak, a multi-task speech-based foundation model designed to
handle children’s unique speech patterns. KidSpeak utilizes a two-stage training strategy,
integrating phonetic insights into the speech encoder, and achieves an average accuracy of
87% across four distinct tasks. The paper also proposes the Flexible and Automatic Speech
Aligner (FASA), which is used to build high-quality annotated datasets in an automated way.

**Strengths:**

1) This paper tackles a crucial question in developing a foundational model for processing
children’s speech, a relatively underexplored area. By integrating large language models
(LLMs) like Whisper and Vicuna and fine-tuning Vicuna with LoRA, the study
demonstrates improved performance over the baseline method.
2) A two-stage training procedure integrates phonetic information into the Whisper speech
encoder, improving downstream performance.

**Weaknesses:**

1) A major limitation of this work is that the proposed KidSpeak model is only compared
against a single baseline (PandaGPT). Numerous multimodal models, such as
ImageBind [1] and NextGPT [2], among others, are available for comparison, along with
robust single-task models (such as wav2vec 2.0, Whisper,  and HuBERT) specifically designed for age classification, gender
classification, automatic speech recognition, and other relevant tasks, whose results
could also be included.

2) The proposed model shows limited novelty, as it primarily relies on existing Whisper
(speech encoder) and Vicuna-based LLMs for its backbone and uses LoRA for
fine-tuning Vicuna - an approach already established in the literature [2]. In contrast, Wu
et al. [2] enhance their multimodal architecture by integrating diffusion-based decoders,
which significantly expands the capabilities of the foundation model.
3) The proposed FASA method is relatively naive, relying primarily on pre- and
post-processing steps. Additionally, Table 4 compares the Montreal Forced Aligner
(MFA) with FASA using only two randomly selected samples, which limits the reliability of
these findings.
4) The presentation and writing in the paper could be improved. For example, usage of
difficult words like "preponderance", "aural", "abridged summary", etc. should be
avoided. The proposed architecture shown in Figure 2 is confusing e.g. how was the text
embedding matrix obtained?

[1] Girdhar, R., El-Nouby, A., Liu, Z., Singh, M., Alwala, K. V., Joulin, A., & Misra, I. (2023).
Imagebind: One embedding space to bind them all. In Proceedings of the IEEE/CVF
Conference on Computer Vision and Pattern Recognition (pp. 15180-15190).

[2] Wu, S., Fei, H., Qu, L., Ji, W., & Chua, T. S. (2024) NExT-GPT: Any-to-Any Multimodal LLM.
In Forty-first International Conference on Machine Learning.

**Questions:**

N/A

---

### Official Review · Reviewer_kaYn · 2024-11-03

**Soundness:** 1
**Presentation:** 2
**Contribution:** 2
**Rating:** 3
**Confidence:** 4

**Summary:**

This paper introduces KidSpeak, a multi-task speech recognition model specifically designed for children's speech. Additionally, it presents the Flexible and Automatic Speech Aligner (FASA), a tool developed to enhance the alignment accuracy between children's speech and corresponding transcriptions.

**Strengths:**

Children’s speech recognition and diagnosis is a critical yet underexplored area in the existing literature.

**Weaknesses:**

1. The presentation of this paper falls short of academic standards, resembling a casual blog post rather than a formal research paper. Lines 82-86 include overly flashy content that detracts from readability and lacks appropriate captions for clarity. Furthermore, the equations on lines 213-215 are unnumbered, which disrupts the structure. The formatting in lines 439-455, featuring diamond-shaped bullet points, italicized beginnings, and inline numbered items with a black background and white text, is visually distracting. Additional typographical errors and inconsistent formatting further detract from the presentation. Figures and tables would benefit from clearer labeling and more precise referencing within the text. Additionally, the writing style shifts between technical and informal, and adopting a more consistent academic tone would improve clarity. Given these issues, the paper may not currently meet the submission standards for ICLR, as it does not adhere to conventional academic writing practices.

2. The forced alignment tool introduced in the paper cannot be considered a novel contribution. The multi-task model essentially adapts existing automatic speech recognition (ASR) and large language model (LLM) frameworks to the domain of children's speech, offering only incremental modifications without substantial innovation over previous work.

3. While the paper reports performance gains from the multi-head Whisper approach, it fails to address the significant computational overhead introduced by the additional decoder. Claiming improved performance from an extra decoder without acknowledging the associated computational costs renders the contribution less meaningful.

**Questions:**

1. What specific innovations does KidSpeak offer that distinguishes it from existing ASR and LLM frameworks?

2. How do the authors address ethical considerations in the collection and use of children's speech data? Are there protocols in place to ensure data privacy and regulatory compliance?

**Details Of Ethics Concerns:**

How do the authors address ethical considerations in the collection and use of children's speech data? Are there protocols in place to ensure data privacy and regulatory compliance?

---

### Official Review · Reviewer_d8WX · 2024-11-04

**Soundness:** 1
**Presentation:** 2
**Contribution:** 2
**Rating:** 3
**Confidence:** 3

**Summary:**

The paper presents approaches to improve automatic annotation of children's speech. First they present KidSpeak a multi-task speech-input foundation model for use with children's speech to yield gender, disorder, ago-group classifications and automatic transcription; this model is trained based on the sequence-to-sequence Whisper model with a dual-decoder (referered to as multi-head) one for orthographic transcription and the other for phonetic transcription. They show significant improvements on TIMIT, a widely used phonetic transcription task, over Whisper. The second contribution is FASA forced alignment tool for generating alignments from noisy speech to generating kids corpus data.

**Strengths:**

The paper is providing advancements in the area of children's speech analysis which is an under-researched area and could use more innovation. Part of the problem is the lack of corpora which this paper alleviates. The results show large improvements over the quoted baselines  1. in phonetic error rate on Timit over Whisper, and 2. a variety of tasks over Panda GPT. FASA is also shown to provide better alignments over human annotators.

**Weaknesses:**

In table 2, of this paper, the Wav2Vec 2.0 number is quoted as 9.7% PER, however in the Wav2Vec paper (https://arxiv.org/pdf/2006.11477v3) it indicates 8.3% and is lead entry in paperswithcode (https://paperswithcode.com/sota/speech-recognition-on-timit). It seems the pre-training with a latent phonetic space as in wav2vec + fine-tuning is enough to yield good results.

In table 4, the computing both WTA and CTA averages (word and char transcription) and then including the inthe multi-task performance average is pretty strange considering they are directly correlated; they can be both reported as a diagnositc, but only one or the other should be incorporated in the average. Its unclear how to interpret these results since PandaGPT is not widely used; how well would humans perform on these tasks or ChatGPT?

On FASA, there are minimal comparison to other approaches and the 13.6% *times* lower WER is hard to take at face value. If the annotators are so bad, how do you even measure this result?

The references are poor. Many references are based on arxiv sources, most appear to be arxiv links for actual peer-reviewed articles, and in this case the citation should note the actual proceedings. Some citations have no source. Two egregious citations are:
- a primary citation for comparable baseline is PandaGPT which is not peer-reviewed
- the Attia 2023 citation has only title and date, no source information. This is the citation for comparison in the primary contribution of "FASA achieves one magnitude lower WER (13.6×)".
While citing from Technical Reports/arxiv is not uncommon, a version and download date needs to be indicated, and peer-reviewed sources are preferred.

Overall, its unclear how impactful this paper is for the general ICLR community given its narrow focus on the domain of children's speech processing.

**Questions:**

What is the size of the Whisper model encoder / decoder used in this work?

Why is there a discrepancy between the Wav2Vec number reported in the paper (9.7%) and the public number (8.3%)?

On FASA, have you looked at other approaches to flexibly aligning to long audio, e.g. https://ieeexplore.ieee.org/document/4960722

If you based your results based on the Wav2Vec model would you have similar results, or would the pre-training negate the dual-decoder improvements?

**Details Of Ethics Concerns:**

No immediate concern with the protocol in the paper, but the paper is using corpora without adequate citation, e.g. ENNI indicates being sources from: (P. Schneider, R. V. Dub´e, and D. Hayward. The edmonton narrative norms instrument, 2005.) This has no source and so unclear whether the children's speech collected for this corpus, or other corpora, were conducted in compliance with FTC's COPPA or other prevailing regulations.

---

### Note · Authors · 2024-11-26

I have read and agree with the venue's withdrawal policy on behalf of myself and my co-authors.